# Estimation of Thermal Properties of Straw-Based Insulating Panels

**DOI:** 10.3390/ma15031073

**Published:** 2022-01-29

**Authors:** Łukasz Czajkowski, Robert Kocewicz, Jerzy Weres, Wiesław Olek

**Affiliations:** 1Faculty of Forestry and Wood Technology, Poznań University of Life Sciences, ul. Wojska Polskiego 28, 60-637 Poznań, Poland; lukasz.czajkowski@mail.up.poznan.pl; 2VestaEco Composites sp. z o.o., ul. Domaniewska 37/2.43, 02-672 Warszawa, Poland; robert.kocewicz@vestaeco.pl; 3Faculty of Information Technology and Visual Communication, Collegium Da Vinci, ul. gen. Tadeusza Kutrzeby 10, 61-719 Poznań, Poland; jerzy.weres@cdv.pl

**Keywords:** bio-based materials, specific heat, thermal conductivity, calorimetric method, inverse modeling

## Abstract

Cereal straw is an environmentally friendly, rapidly renewable, and sustainable raw material for manufacturing insulating panels for building engineering. Credible data on thermal properties of insulating panels are crucial for appropriate and accurate design of building envelopes. The objective of the study was to determine and validate thermal properties of the panels made of cereal straw. Specific heat was measured with the calorimetric method. Thermal conductivity was determined with the inverse method and Isomet 2114 instrument, respectively. Both approaches accounted for the temperature influence. The specific heat of the panels was as high as 1600 J/(kg·K), while the thermal conductivity varied in the range from 0.025 to 0.075 W/(m·K) depending on the applied experimental method. The studied properties were validated and their credibility was assessed. High accuracy of heat transfer modeling was obtained for the properties measured with the calorimetric method and identified with inverse modeling.

## 1. Introduction

The climate change agenda induces development of low energy and passive building constructions in which biomaterials are used. Currently, the majority of building materials are produced from nonrenewable resources by using fossil fuels. The production processes of the building materials significantly increase the carbon footprint of buildings, also due to high energy consumption. The application of natural, renewable, sustainable, and environmentally friendly materials such as wood or agricultural byproducts significantly reduces the amount of waste materials and environmental pollution [1,2,3,4]. The most popular natural building materials are wood and annual plants being used as structural and/or insulation materials. Cereal straw, being the byproduct during grain harvesting, is considered one of the most eco-friendly and recyclable raw materials. The straw properties, widespread availability, and renewability make the material more and more frequently applied in building engineering [5,6,7,8].

The analysis of the thermal properties of raw cereal straw depicted high values of porosity and specific heat [9,10,11]. It suggests potential usefulness of cereal straw as a material for producing insulation composites. The insulating panels made of annual plants are characterized by low values of thermal conductivity coefficient similar to the ones reported for synthetic materials and rock wool [12,13,14]. Particles of annual plants are also applied as filling materials in concrete constructions in order to improve its thermal properties [15,16].

The energy consumption in buildings for heating is regulated by law and novel solutions meet more and more restrictive demands on insulating properties of barrier envelopes depending on the overall heat-transfer coefficient. The values of the coefficient primary result from thermal conductivity and thickness of the applied insulating material. Credible data on the thermal properties of the materials are required for correct design of barrier envelopes, especially in the case of advanced low-energy and passive buildings [17].

The thermal properties of insulating materials are most often determined with the use of meters based on the steady-state method, e.g., guarded hot plate meters and heat flow meters. The implied method causes many limits for the application of the meters. The most important one excludes the use of the meters for examining anisotropic materials. Another important limitation is related to the problem of the contact resistance between a tested sample and a plate or a sensor of a meter. The imperfect contact significantly underrates the measured values of thermal conductivity [18]. The insulating materials are characterized by low density and high porosity. The pores are filled with air, characterized by thermal conductivity of ca. 0.025 W/(m·K) [19]. Therefore, the presence of air in pores significantly improves the insulating properties.

Domínguez-Muñoz et al. [20] revealed that the thermal conductivity of insulating materials increases with a density decrease below ca. 30 kg/m^3^. The lowest and practically constant values of the thermal conductivity were found for the materials characterized by density varying from 30 to 60 kg/m^3^. The density increase above 60 kg/m^3^ caused a gradual and distinct increase of thermal conductivity, which was explained by the porosity decrease. A similar relation was found by Csanády et al. [21] for insulating panels made of straw. The values of the thermal conductivity were determined with the guarded hot plate method and related to density of the panels. The minimum value of the thermal conductivity was found for the density of 120 kg/m^3^. The thermal conductivity of the investigated materials was attributed to three different mechanisms of heat transfer, i.e., conduction in stems of straw, conduction in air, and radiation. The convective heat transfer in air was neglected.

The natural bio-based insulating materials contain open pores which are also filled with air. Unfortunately, during heat transfer in such materials, the air flows within the open porous structure. Such a process affects insulating properties, especially for high differences of temperature within the material at the near-surface layer, being in contact with a plate or a sensor. The phenomenon of air flow within tested materials is not accounted for by the meters [22].

Another group of meters is grounded on the transient method [23]. The meters can be used for the thermal conductivity measurements performed for a wide range of temperature values. Moreover, the transient meters can indirectly account for the influence of air flow in the porous structure of bio-based materials as well as changes in air density on the intensity of heat transfer during the measurements.

The objective of the study was to determine thermal properties of insulating panels made of cereal straw. Two different methods were used and compared for measurements in the temperature range of 10–50 °C. The validation procedure was applied to quantify the credibility of the applied measurement methods.

## 2. Materials and Methods

### 2.1. Insulating Panels

The commercial insulating panels were the focus of the study. The panels were manufactured by VestaEco company (Warsaw, Poland) from a mix of rye and triticale straw. The straw was ground to a particle size ranging from 10 to 50 mm of length. The obtained particles were firstly hydrothermally and chemically treated at temperature lower than 50 °C and then subjected to defibering with the use of the DefibraTech 1.0 technology. The water excess was squeezed out of the defibered mass then fluffed and dried to a moisture content of ca. 10%. The dried fibers were blended with polymeric 4,4′-diphenylmethane diisocyanate (pMDI) resin with the resin load of ca. 4%. The mattress was formed and pressed at temperature of 165 °C, and the one-layer panels of thickness of 40 mm were produced. The target density of the panels was 210 kg/m^3^.

### 2.2. Specific Heat and Bulk Density Determination

The specific heat of the examined panels was measured with a water calorimeter dedicated to the bio-based materials, characterized with low density and heat capacity as well as high hygroscopicity. In order to design and construct the calorimeter, the steady-state heat balance equation was first formulated and the analysis of the absolute error (uncertainty) of the specific heat determination was carried out with the use of the total differential method [24].

The round sheets of oven-dry panels were stacked into samples in the shape of cylinders of a diameter of 80 mm and a height of 140 mm. The samples were placed in a heat shrinkable membrane in order to prevent changes of moisture content of the examined panels. The temperature distribution in the panels was controlled during the experiments by two type K thermocouples mounted in the center and at the surface of the samples. The equilibrium temperature of the calorimetric system was found with the use of type J thermocouple. The detailed description of designing and constructing the calorimeter as well as the procedure of the specific heat determination was provided by Czajkowski et al. [24].

The specific heat measurements were followed by the bulk density determination. The thermocouples were dismounted from the samples and the panels were again covered with heat shrinkable membrane. This was made because of volume determination by water displacement in a calibrated cylinder. The mass and volume of heat shrinkable membrane were subtracted from the readings of volume and mass of samples. The bulk density of the examined panels was calculated as the ratio of mass and volume.

### 2.3. Thermal Conductivity Determination with the Transient Method (Isomet 2114)

The transient methods for the thermal conductivity determination are characterized by a relatively short duration of the measurements as well as simple handling during experiments, having been made for samples of different shapes. Therefore, the measurements were done with the Isomet 2114 instrument (Applied Precision s.r.o., Bratislava, Slovakia) which is a commercial hand-held measuring system operating on the basis of transient heat conduction. The instrument is usually used for determining thermal properties of different isotropic materials including insulating ones. The measuring principle of the Isomet 2114 is based on transient plane source (TPS) method. The surface measuring sensor was in the direct heat contact with a face of a tested sample.

The samples in a shape of rectangular prisms of the dimensions of 150·150·40 mm were firstly dried to the absolutely dry state. The dried samples were cooled down and then wrapped with low density polyethylene (LDPE) foil in order to prevent changes of moisture content of the investigated material. A portion of foil was removed from a sample to enable the direct contact of a measurement probe with a surface of the material. A sample with the probe was placed in the climate chamber. The temperature of the experimental system was firstly equilibrated for 60 min before starting measurements. The experiments were done for three levels of temperature, i.e., 10, 30, and 50 °C. The instrument measured the thermal conductivity and related it to the average testing temperature. The duration of a single measurement was ca. 15 min. For a given temperature level, the measurements were done for six samples with six repetitions for each sample.

### 2.4. Thermal Conductivity Identification with the Inverse Modeling

The inverse identification of thermal conductivity was applied for determining thermal conductivity of the panels. The identification was based on the approach proposed by Weres and Olek [25] and Weres et al. [26]. It exploited measured responses of the examined empirical system, i.e., temperature values in time as measured in selected locations of the investigated material. In order to collect the responses, the transient heat transfer experiments had to be performed. The experimental material was firstly oven dried in a laboratory dryer and formed into cube-shaped samples of the dimensions of 100 × 100 × 100 mm. A set of four type K thermocouples was mounted in each sample and coded as #1, #2, #4, and #6. The locations of the thermocouples are listed in Table 1. The origin of the ortho-Cartesian system of coordinates was located at the front bottom left corner of a sample. Each sample was also equipped with three additional thermocouples with codes #3, #5, and #7, which were installed at surfaces of the samples in order to register temperature values at all faces of the samples. In order to get a good contact of the thermocouples #3, #5, and #7 with the faces of the samples, a self-adhesive aluminum foil was used. The application of the foil reduced the contact resistance and thus improved the accuracy of temperature measurements at the faces of the samples.

A single transient heat transfer experiment comprised two stages, i.e., (a) heating of a sample in order to obtain the uniform spatial distribution of temperature in a sample (the initial condition) and (b) cooling of the same sample. The temperature values in time were registered every 60 s in the cooling stage. The recorded values were the input data for the thermal conductivity identification with the inverse method. The validation of the identified thermal conductivity was made with the use of other sets of data registered during transient heat transfer experiments.

The mathematical structural model of transient three-dimensional heat conduction was applied for the identification. The model accounted for the temperature dependence of the thermal conductivity on temperature. It also assumed the uniform initial distribution of temperature within samples (the initial condition) and the Dirichlet boundary condition as the temperature values at all faces of the samples were measured during the transient experiments. The model was composed of the following quasi-linear parabolic partial differential equation:(1)$$c\rho \;\frac{\partial \;t}{\partial \tau }\;=\frac{\partial }{\partial x_{i}}\;\left(k\;\frac{\partial t}{\partial x_{i}}\right)\;,\;\;\;\;\;\;\;\;\;\;\;\;\;\;\;\;\;\;\;\;\left(x_{i},\;\tau \right)\in \Omega \times (0,\;\tau_{F}]\;$$
with the initial condition
(2)$$t\;\left(x_{i},0\right)=t_{0}\left(x_{i}\right)\;,\;\;\;\;\;\;\;\;\;\;\;\;\;\;\;\;\;\;\;\;\;\;\;\;\;\;\;\left(x_{i}\right)\in \Omega$$
and the first kind boundary condition
(3)$$t\;\left(x_{i},\tau \right)=t_{s}\left(x_{i}\right)\;,\;\;\;\;\;\;\;\;\;\;\;\;\;\;\;\;\;\;\;\;\;\;\;\;\;\;\left(x_{i},\tau \right)\in \partial \Omega^{I}\times (0,\tau_{F}]\;$$
with *i* = 1, 2, 3

where *c*, J/(kg·K)—specific heat; *k,* W/(m·K)—thermal conductivity; *t,* °C—temperature; *t*_0_, °C—initial temperature; *t_s_*, °C—temperature at the boundary; *x_i_*, m—coordinates of a point in the ortho-Cartesian system of coordinates; *ρ*, kg/m^3^—density; *τ*, s—time; Ω, m^3^—domain of the body examined in the three-dimensional Euclidean space; *τ_F_*, s—final time of the heat conduction process; ∂Ω*^I^*, m^2^—boundary of the domain for the first kind boundary condition.

The finite element method was used to develop the operational form of the model given by Equations (1)–(3). The approximation of the geometrical domain was made with 3D space elements (rectangular prisms), while the time domain was approximated with the absolutely stable two-point recurrence scheme. Moreover, the iteration procedure was used to deal with the quasi-linearity of equations at each time step. The final form of the operational model was obtained as the sets of algebraic equations, and the nodal values of temperature at selected time instants were the values sought [27].

The results of solving the direct problem of heat conduction as described by Equations (1)–(3) were compared to the empirical data collected during the transient heat transfer experiments (the measured responses of the examined empirical system). The comparison was quantified by calculating the objective function defined as
(4)$$S={\sum_{i=1}^{NT}w_{i}\;\left[t_{exp}\left(\tau_{i}\right)-t_{pred}\left(\tau_{i}\right)\right]}^{2}$$
where *w_i_*—weight function; *t_exp_*, *t_pred_*—temperature values at selected nodal locations as measured and predicted, respectively; and *NT*—number of time instants.

The optimization procedure was used to determine the minimum of the objective function with respect to the mathematical model coefficients subject to estimation. The procedure was based on the trust region algorithm as combined with the secant-updating quasi-Newton procedure to approximate the Hessian (the BFGS update).

## 3. Results

The measurements of specific heat and bulk density of the panels were the first analyses made in the present study. This was due to serious limitations of the inverse identification procedure being applied for determining coefficients of the heat conduction model [28]. It was found that the simultaneous identification of specific heat and thermal conductivity invoked finding the infinite number of pairs of the coefficients. Kim et al. [29] recommended to determine specific heat firstly, and then to exploit the measured values as input data for the thermal conductivity identification with the inverse method. Borges et al. [30] applied a sensitivity analysis for determining a dependence of the identified coefficients during the inverse analysis of heat transfer problems. It was concluded that the linear dependence of two or more coefficients made it impossible for the simultaneous estimation of the coefficients.

### 3.1. Density and Specific Heat

The analyzed insulation panels represented low volumetric specific heat capacity. Therefore, the calorimetric measurements were characterized by a relatively small amount of heat being released by a sample during experiments. This implied the requirement for preparing samples of relatively high mass, in our case ca. 180 g. This resulted from the already performed error analysis [24] that the minimum temperature increase of the calorimetric system (Δ*T*) should be equal to 1.5 K. The initial equilibrium temperature of the calorimetric system (*t_e_*) was always equal to ca. 20 °C (Table 2). It forced us to heat the samples to the initial temperature (*t_is_*) of ca. 100 °C. The specific heat values were calculated from the transformed heat balance equation as derived by Czajkowski et al. [24] and presented in Table 2 as well as supplemented by the results of the oven dry density of the panels and calculated values of volumetric specific heat capacity. The standard deviation of the specific heat measurements was lower than 8 J/(kg·K) as the experimental material was carefully prepared and high repeatability of the experimental results was enabled.

The obtained mean value of the specific heat was 1678 J/(kg·K) while density was 211 kg/m^3^ (Table 2). This resulted in a volumetric specific heat capacity of 0.3538 MJ/(m^3^·K). The direct comparison of the results of the specific heat and density is difficult as the properties of insulating panels made of cereal straw are rarely reported. Palumbo et al. [13] analyzed six different bio-based insulating boards made of corn pith, barley straw, hemp fibers, a mixture of hemp hurds and lime, wood wool, and wood fibers. The Quickline-30 Electronic Thermal Properties Analyser based on the transient hot-wire method was applied for measuring the thermal conductivity and thermal diffusivity of the examined boards. The measured properties were used for calculating volumetric specific heat capacity of the boards. The reported values of the heat capacity for the oven-dry state varied from 0.0384 to 0.1612 MJ/(m^3^·K) for boards made of corn pith and wood fibers, respectively. The corresponding values of oven-dry density were ranging from 48.1 to 212.2 kg/m^3^. The volumetric specific heat capacity registered in the present study was practically two times higher as compared to the highest value reported by Palumbo et al. [13]. This was probably due to the applied transient hot-wire method for investigating boards. The method is the most suitable for investigations on liquids and gases and it has serious limitations as applied to solids. The limitations imply measurement uncertainties resulting mainly from the existence of the contact resistance between a sensor and a sample surface, determination of the amount of heat emitted by a sensor, assumption of the infinite length, and legible diameter of the wire. Bejzak and Zvizdić [31] listed and analyzed several factors influencing measurement uncertainties of the method. It was also demonstrated that the transient hot-wire method revealed higher measurement uncertainties as compared to the steady-state guarded hot-plate method.

Hussain et al. [32] reported manufacturing and testing water-resistant hemp shiv-based composites for potential use as thermal insulating materials. The composites differed in the type and amount of the applied binders which resulted in variation of the bulk density from 175 to 240 kg/m^3^. The thermal properties of the manufactured hemp composites were measured with the Isomet 2114 instrument operating according to the TPS method being similar in its principles to the transient hot-wire method. The measured thermal properties and the bulk density were used to calculate the specific heat of the composites and exceptionally low values ranging from 763 to 1050 J/(kg·K) were found. It can be again explained by the measurement uncertainties resulting from the applied experimental method and the measurements being done for unknown moisture content.

The similar approach was used by Liuzzi et al. [5] to determine the thermal properties of insulating panels made of barley straw fibers and olive tree wastes. Again, the Isomet 2114 was used to measure thermal properties, here thermal conductivity, thermal diffusivity, and volumetric heat capacity. The latter property was used together with bulk density to calculate specific heat of the panels. The bulk density of the panels made of barley straw and olive waste was 152 and 235 kg/m^3^, respectively. The indirectly evaluated values of the specific heat were equal to 1010 and 1111 J/(kg·K) for barley straw and olive wastes panels, respectively. The same apparatus was applied as in the previous study. However, the measurements were made for the dry materials. The values were again very low.

### 3.2. Thermal Conductivity Measured with Isomet 2114

In contrast to the inverse identification, the thermal conductivity measurements with the Isomet 2114 were performed independently on the information on density and specific heat. The measurements were made for three levels of temperature, i.e., 10, 30, and 50 °C and with a use of a set of six samples in a shape of rectangular prisms as described in the Section 2. The number of repetitions for a given temperature level and a sample was six. This resulted in a total number of observations of 108. The measured thermal conductivity values were parametrized with an empirical linear model giving the following relation
*k* = 0.04459 + 0.0002767 · *t*(5)
where *k*, W/(m·K)—thermal conductivity; *t*, °C—temperature.

The measured discrete data and the fitted linear model are depicted in Figure 1. The obtained values were in accordance with the observations reported by Hussain et al. [32] who found the thermal conductivity values varying from 0.052 to 0.057 W/(m·K) for the insulating composites made of hemp shiv. Similar values were obtained by Liuzzi et al. [5] for insulating panels manufactured from barley straw and olive tree wastes with values of 0.058 and 0.062 W/(m·K), respectively.

### 3.3. Thermal Conductivity Identified with the Inverse Method

The inverse identification of the thermal conductivity also enabled accounting for the thermal conductivity dependence on temperature. Two options of the temperature influence were considered. The first one assumed linear dependency of the thermal conductivity on temperature and resulted in the following relation *k* = −0.0155 + 0.001346 · *t* where thermal conductivity (*k*) was expressed in W/(m·K) and temperature (*t*) varied from 30 to 60 °C. The second option postulated that there is no influence of temperature on the identified property and it turned out that the thermal conductivity was equal to 0.0744 W/(m·K). The thermal conductivity values identified with the inverse method and measured with Isomet 2114 are depicted in Figure 2.

### 3.4. Thermal Properties Validation

The measured and identified thermal properties of the insulating panels were validated by comparing the results of transient heat transfer modeling to the set of empirical data, i.e., the measured temperature values in time in selected locations. This set of empirical data was only used for validation, and not in the process of identification. The validation was made for density and specific heat capacity measured with the calorimetric method as well as for identified and measured thermal conductivity. This resulted in three options of the validation coming up from the identification of thermal conductivity independent and dependent on temperature as well as thermal conductivity measurements with Isomet 2114.

The performed validation was quantified by two errors defined by Olek et al. [33], i.e., the local in time relative error *e*_1_:(6)$$e_{1}\left(x_{i},\tau_{j}\right)=100\frac{\left|\;t_{\exp}\left(x_{i},\tau_{j}\right)-t\left(x_{i},\tau_{j}\right)\;\right|}{t_{\exp}\left(x_{i},\tau_{j}\right)},\;\;\;\;\;\;\;i\;=\;1,\ldots ,NS\;,\;\;\;j\;=\;1,\ldots ,NT$$
where *t*_exp_, °C—temperature; *NS*—number of thermocouples; *NT*—number of time intervals;

and the global in time relative error *e*_2_:(7)$$e_{2}\left(x_{i}\right)=100\frac{\sqrt{{\sum_{j=1}^{NT}\;\left[t_{\exp}\left(x_{i},\tau_{j}\right)-t\left(x_{i},\tau_{j}\right)\right]}^{2}}}{\sqrt{\sum_{j=1}^{NT}\;t_{\exp}{\left(x_{i},\tau_{j}\right)}^{2}}},\;\;\;i\;=\;1,\ldots ,NS$$

The examples of the performed validation are depicted in Figure 3 and Figure 4. The top plots presented comparisons of the results of the three options of modeling to the empirical data, while the bottom plots depicted the evolution of the local in time relative error *e*_1_. The global in time relative error *e*_2_ was calculated for all thermocouples mounted inside a sample (thermocouples #1, #2, #4, and #6). The obtained values of *e*_2_ were presented in Table 3.

The global in time relative error (*e*_2_) values were lower than 2% for all locations of thermocouples and identified thermal properties accounting for the thermal conductivity dependency on temperature (Table 2). The two times higher values of the *e*_2_ error, i.e., ca. 4%, were found for the identification option assuming no influence of temperature on thermal conductivity. In the case of the validation of the thermal conductivity measured with Isomet 2114, the *e*_2_ error varied from ca. 3% to 5% depending on the thermocouple location. The analysis of the local in time relative error *e*_1_ (bottom plots in Figure 3 and Figure 4) clearly showed that the maximum of the *e*_1_ error was less than 3% for the thermal conductivity accounting for the temperature influence and identified with the inverse method. The maximum value found for the validation of the thermal conductivity measured with Isomet 2114 was ca. 8%. The error analysis confirmed significantly higher accuracy of the identification accounting for the temperature dependency of the thermal conductivity as compared to the values measured with Isomet 2114.

## 4. Conclusions

The water calorimeter was effectively used for determining the specific heat of straw-based insulating panels. The exploited calorimeter guaranteed high repeatability and accuracy of specific heat measurements, i.e., the standard deviation of the measurements was lower than 8 J/(kg·K). The investigated insulating panels were characterized by high values of the specific heat of over 1600 J/(kg·K), which resulted in finding volumetric specific heat capacity of ca. 0.35 MJ/(m3·K) despite a low density of ca. 210 kg/m3.The design of the experiment for measuring the thermal conductivity with Isomet 2114 instrument enabled accounting for temperature influence on the property. The discrete values of the thermal conductivity were parametrized with the linear function. The obtained results of the thermal conductivity were similar to the findings obtained for the insulating composites made of hemp shiv and insulating panels manufactured from barley straw and olive tree wastes.The two options of the inverse identification were highly effective in determining the thermal conductivity. However, the identification option assuming the linear dependence of the thermal conductivity on temperature revealed higher accuracy as compared to the option which postulated that there is no influence of temperature on the identified property. It clearly shows the importance of more accurate modeling by accounting for the temperature influence on the thermal conductivity.The validation procedure enabled quantification of the credibility of the applied experimental methods for determining thermal properties of straw-based insulating panels. It was clearly depicted that accurate modeling of heat transfer in the insulating panels is possible when using the specific heat as determined by the calorimetric method together with the thermal conductivity identified with the inverse modeling or measured by the applied instrument. However, it was presented that higher accuracy is obtained when using the identified thermal conductivity accounting for the linear dependency on temperature (the local in time relative error e1 did not exceed 2% and the global in time relative error e2 varied from ca. 3% to 5% for this option of the identification).

## Figures and Tables

**Figure 1 materials-15-01073-f001:**
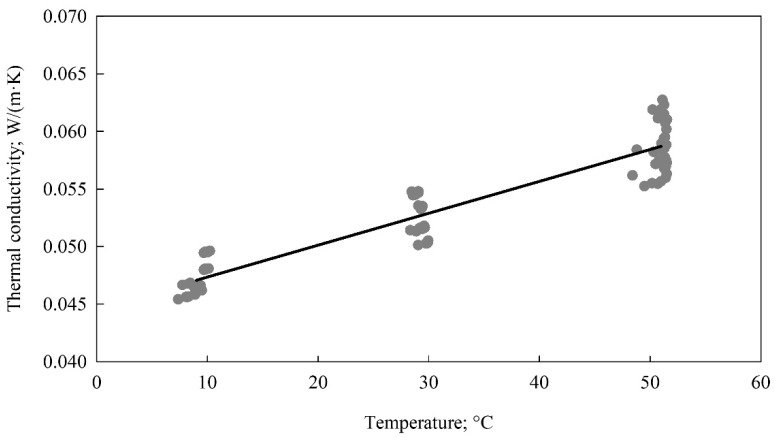
Thermal conductivity measured with Isomet 2114 (dots—individual observations, solid line—empirical model).

**Figure 2 materials-15-01073-f002:**
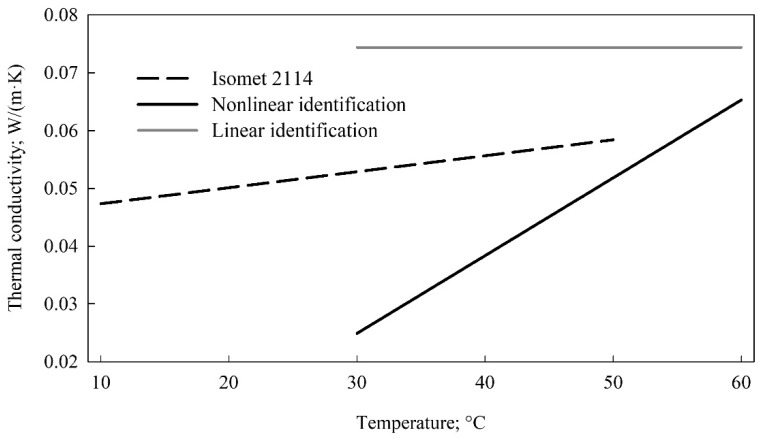
Comparison of thermal conductivity values identified with the inverse method and measured with Isomet 2114.

**Figure 3 materials-15-01073-f003:**
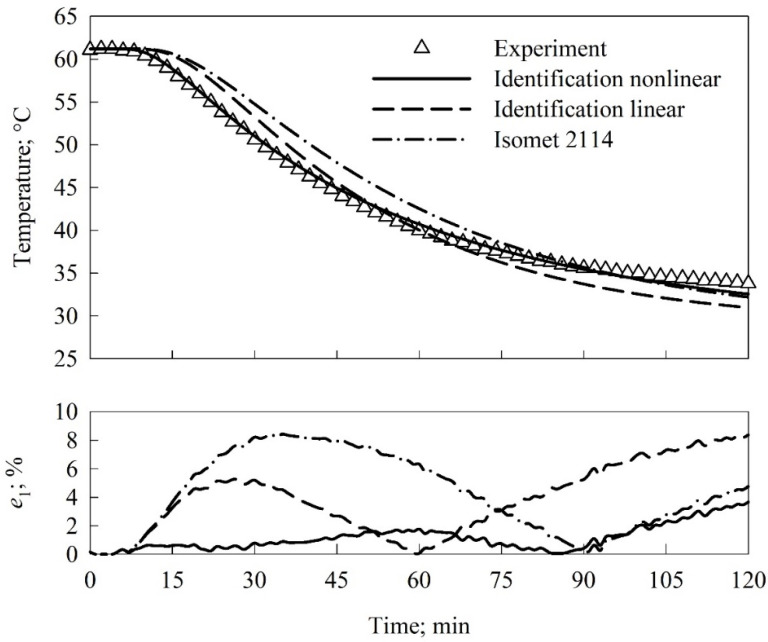
Predicted temperature values as functions of time for measured and identified thermal properties compared to experimental data (upper plot) and the relative error *e*_1_ of modeling (bottom plot). Thermocouple #1.

**Figure 4 materials-15-01073-f004:**
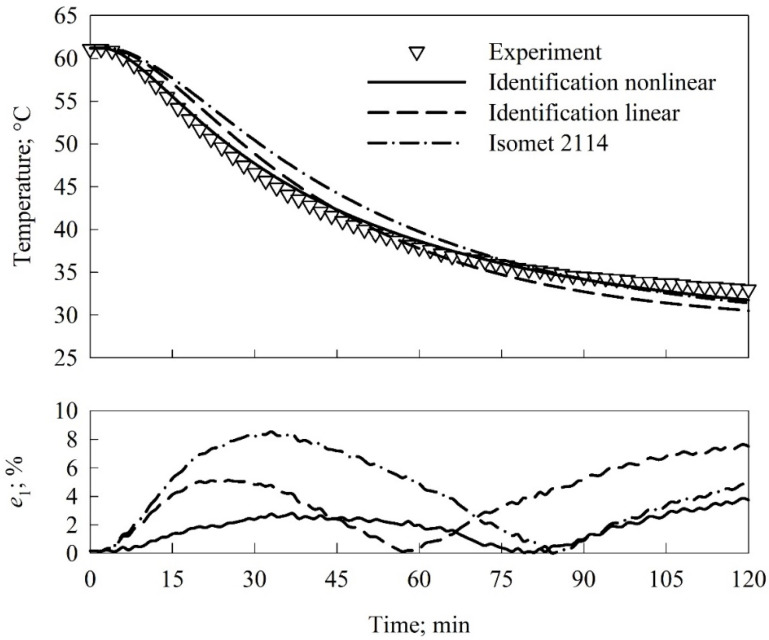
Predicted temperature values as functions of time for measured and identified thermal properties compared to experimental data (upper plot) and the relative error *e*_1_ of modeling (bottom plot). Thermocouple #4.

**Table 1 materials-15-01073-t001:** Coordinates of the thermocouple locations in mm.

Coordinates	Thermocouple Location						
	#1	#2	#3	#4	#5	#6	#7
*x* _1_	50	50	50	50	50	75	100
*x* _2_	50	25	0	50	50	50	50
*x* _3_	50	50	50	75	100	50	50

Note: The thermocouples #3, #5, and #7 were mounted on the faces of the samples and provided the data for the boundary conditions.

**Table 2 materials-15-01073-t002:** Individual observations and mean values of specific heat, density, and volumetric specific heat capacity of the examined panels.

Observation	Sample Initial Temperature *t_is_*; °C	Initial Equilibrium Temperature of the Calorimetric System *t_e_*; °C	Increase of Temperature Δ*T*; K	Specific Heat *c*; J/(kg·K)	Density *ρ*; kg/m^3^	Volumetric Specific Heat Capacity *c*·*ρ*;MJ/(m^3^·K)
#1	98.0	18.56	1.41	1672	210	0.3511
#2	99.6	18.41	1.46	1668	203	0.3386
#3	98.2	17.65	1.46	1682	211	0.3549
#4	99.4	18.02	1.46	1683	212	0.3568
#5	99.2	18.34	1.46	1686	218	0.3675
**Mean value**	-	-	-	**1678**	**211**	**0.3538**

**Table 3 materials-15-01073-t003:** The values of the *e*_2_ error for the analyzed options of validation.

Thermocouple Number	Option of Validation		
	Nonlinear Identification	Linear Identification	Isomet 2114 Measurements
#1	1.21	4.09	5.19
#2	1.42	3.73	3.65
#4	1.88	4.05	5.10
#6	1.98	3.71	2.78

## Data Availability

Not applicable.

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
