# Peer review of "Estimation of Thermal Properties of Straw-Based Insulating Panels"

_materials, 2022, doi:10.3390/ma15031073_

Round 1

Reviewer 1 Report

The paper presents an experimental study on the thermal properties, e.g., specific heat and thermal conductivity of straw-based insulating panels as construction materials. Commercial insulation panels were tested, and both transient method and inverse modeling method were used for the thermal conductivity measurements. The results are interesting to the field. Here are some comments to improve the paper.

  1. The writing of the paper could be improved. Please try to structure the paragraphs in a clearer manner and avoid piling irrelevant sentences together.
  2. On page 1, please add the references to justify the statements of “the most popular natural building materials” and “the most eco-friendly and recyclable raw materials.” I suspect that they are only one of the most.
  3. On page 2, the 3 – 5 paragraphs are not clear. Please revise and add appropriate references.
  4. On page 3, please revise the unit of temperature.
  5. On page 4, please provide more details about how the thermocouples #5 – 7 were installed on the surface. The methods of attaching thermocouples on surfaces can impact the accuracy of the measurement. Please elaborate.
  6. In section 2, the ways to present the labels are not consistent. Please provide units for all the labels in the equations.
  7. On page 6, what are the ‘several factors influencing measurement uncertainties’ analyzed by Bejzak and Zvizdic? How were these factors addressed in the measurements in this paper?
  8. On page 7, about equation (5), why is a linear model used? The figure only provides three temperature measurements, which couldn’t justify a linear model.
  9. On figure 2, why are the thermal conductivities so different? It is not intuitive that such different thermal conductivity values could lead to very similar predictions in figures 3 and 4. Please elaborate on this.

Author Response

Answers to comments of the Reviewer #1:

Comment #1. The writing of the paper could be improved. Please try to structure the paragraphs in a clearer manner and avoid piling irrelevant sentences together.

Answer. The text of the paper was carefully analyzed. The fragments being recognized as requiring some improvement were corrected.

Comment #2. On page 1, please add the references to justify the statements of “the most popular natural building materials” and “the most eco-friendly and recyclable raw materials.” I suspect that they are only one of the most.

Answer. We agree with the remark. Definitely cereal straw is just one of “the most eco-friendly and recyclable raw material” being used in building engineering. It is because timber and wood-based panels play important role in this area. Therefore, we revised our paper in order to be more precise. On the other side we already mentioned in the paper that wood and annual plants are the most popular natural building materials.

Comment #3. On page 2, the 3 – 5 paragraphs are not clear. Please revise and add appropriate references.

Answer. In order to be more specific we added the appropriate references concerning the structure of the natural bio-based insulating materials, the meters being grounded on the transient method, and the thermal conductivity of air.
Added references:
Çengel, Y.A.; Ghajar, A.J. Heat and mass transfer: fundamentals & applications. Fifth Edition. McGraw-Hill, New York, 2015
Ferroukhi, M.Y.; Abahri, K.; Belarbi, R.; Limam, K.; Nouviaire, A. Experimental validation of coupled heat, air and moisture transfer modeling in multilayer building components. Heat Mass Transf. 2016, 52, 2257-2269. DOI: 10.1007/s00231-015-1740-y
Palacios, A.; Cong, L.; Navarro, M.E.; Ding, Y.; Barreneche, C. Thermal conductivity measurements techniques for char-acterizing thermal energy storage materials – A review. Renew. Sust. Energ. Rev. 2019, 108, 32-52. DOI: 10.1016/j.rser.2019.03.020

Comment #4. On page 3, please revise the unit of temperature.

Answer. We would like to thank for pointing out the problem. The unit is corrected in the revised version of the paper.

Comment #5. On page 4, please provide more details about how the thermocouples #5 – 7 were installed on the surface. The methods of attaching thermocouples on surfaces can impact the accuracy of the measurement. Please elaborate.

Answer. The thermocouples #3, #5 and #7 were mounted at the faces of the samples. The thermocouple #6 was always installed inside the samples (please refer to locations listed in Table 1). In order to get a good contact of the thermocouples #3, #5 and #7 with the faces of the samples we used self-adhesive aluminum foil. The application of the foil reduced the contact resistance and thus improved the accuracy of temperature measurements at the faces of the samples. Moreover, the positions of all thermocouples were checked and corrected in the revised version of the paper.

Comment #6. In section 2, the ways to present the labels are not consistent. Please provide units for all the labels in the equations.

Answer. All physical quantities (symbols) in equations (1)-(7) are explained in the paper. The quantities are also supplemented with units. The labels are used to number thermocouples and observations when measuring specific heat and density. Therefore, the remark is unclear.

Comment #7. On page 6, what are the ‘several factors influencing measurement uncertainties’ analyzed by Bejzak and Zvizdic? How were these factors addressed in the measurements in this paper?

Answer. Bejzak and Zvizdić (2011) made detailed analysis of factors influencing measurement uncertainties of thermal properties when applying meters operating on the principle of the transient hot wire method and guarded hot plate method. In the case of the transient hot wire method the analyzed factors were diameter and length of the wire, linearized electrical resistance temperature coefficient of the wire, electrical resistance of the wire, measurement uncertainty of the voltmeter, measurement uncertainty of the ammeter. The practically the same factors had to influence the uncertainty of the measurements being done with Isomet 2114 which is based on the transient plane source method. Thus, the observed values varied greatly at the same temperature level. It is the most important disadvantage of the employed method in Isomet 2114 when using the meter for natural bio-based insulating materials.

Comment #8. On page 7, about equation (5), why is a linear model used? The figure only provides three temperature measurements, which couldn’t justify a linear model.

Answer. The number of the independent variables (i.e. temperature levels) were equal to 3. Therefore, the linear model is the only justified empirical model to describe the influence of temperature on the thermal conductivity.

Comment #9. On figure 2, why are the thermal conductivities so different? It is not intuitive that such different thermal conductivity values could lead to very similar predictions in figures 3 and 4. Please elaborate on this.

Answer. The differences in the thermal conductivity values result from the applied different methods for determining the property. The limitations of the hot wire method and the transient plane source method (as applied Isomet 2114) were already discussed in Results (page 6). The performed error analysis supplementing the validation clearly revealed that much better prediction was always found for the thermal conductivity values as identified with the nonlinear inverse modeling. The prediction differences were quantified with the local in time relative error e1 (Figures 3 and 4) and the global in time relative error e2 (Table 3).

Reviewer 2 Report

The present analysis is a contribution to the general focus of exploring new alternative and low environmental impact materials for ecological transition in the modern construction sector. In particular, the main question addressed by the authors concerns the way thermal properties of bio-based insulating panels can be estimated from an experimental viewpoint. These panels are straw-based ones.
This paper provides an interesting experimental approach for obtaining thermal conductivity measurements of this material. The article is well written and is also well constructed. My comments are as follows:
1. For a better understanding by the reader, a photograph of the different samples should be added.
2. Position of the thermocouple 6 has to be corrected in Table 1.
3. Commercial straw-based panels are insulating ones, mainly used in indoor environment at room temperatures (around 20°C). Linear and non linear identifications are built for temperatures ranging from 30°C to 60°C while Isomet 2114 has been used for temperatures in the range 10°C-50°C. What comparison can you make between these analyses and the measurements with the Isomet 2114 at indoor temperatures?
In conclusion, once minor revisions are made, I recommend this article for publication.

Author Response

Answers to comments of the Reviewer #2:

Comment #1. For a better understanding by the reader, a photograph of the different samples should be added.

Answer. We are afraid that a photograph of the examined material would not provide valuable information. Therefore, we made a decision for not attaching such a picture into the paper.

Comment #2. Position of the thermocouple 6 has to be corrected in Table 1.

Answer. We would like to thank for pointing out the problem. As considers the positions of the thermocouple #6 and the other thermocouples we checked it and corrected in the revised version of the paper.

Comment #3. Commercial straw-based panels are insulating ones, mainly used in indoor environment at room temperatures (around 20°C). Linear and non linear identifications are built for temperatures ranging from 30°C to 60°C while Isomet 2114 has been used for temperatures in the range 10°C-50°C. What comparison can you make between these analyses and the measurements with the Isomet 2114 at indoor temperatures?

Answer. We agree with the reviewer’s comment. We decided to deal with the problem indirectly, following the idea of validating consequences of the hypothesis instead of the hypothesis itself. That idea is known and accepted on the grounds of the methodology of empirical sciences. In the case presented in the paper any comparison cannot be made for 20ºC, as the values appropriate for the inverse problem identification are inaccessible. Therefore we decided to perform an indirect comparison by determining effects of the obtained thermal conductivity values on modeling of the heat transfer, and thus to make an indirect validation.

Reviewer 3 Report

Cereal straw, as an environmentally friendly and sustainable raw material for manufacturing insulating panels, is extremely promising in building engineering field. In this work, thermal properties of commercial insulating panels made of rye and triticale straw were particularly studied. Specific heat and thermal conductivity were measured with the calorimetric method, the inverse method and Isomet 2114 instrument, respectively. This paper can be published although some revisions are required, and specific comments are as follows:

  1. It is recommended to add some recent references.
  2. It would be better to interpret the reason why the values of specific heat, density and volumetric specific heat capacity of observation 1 were below other observation samples (table 1).
  3. Could you carefully explain why individual observation values of thermal conductivity measured with Isomet 2114 varied greatly at the same temperature level (fig. 1).
  4. It would be more coherent to make a comparison between fig. 3 and fig. 4.
  5. Please present data of different thermocouples, such as thermocouple number #3 and #5, to make the table more comprehensive (table 3).

Author Response

Answers to comments of the Reviewer #3:

Comment #1. It is recommended to add some recent references.

Answer. Some recent references were added to the revised version of the paper. The number of cited recent works (i.e. published in years 2019-2022) increased from 8 references (29% of all cited works in originally submitted paper) to 13 references (39% of all cited works in revised version of the paper).

Comment #2. It would be better to interpret the reason why the values of specific heat, density and volumetric specific heat capacity of observation 1 were below other observation samples (table 1).

Answer. The results on specific heat, density and volumetric specific heat capacity are presented in Table 2 and not in Table 1 as it was suggested in the review. Moreover, the values of for the observation #1 are only insignificantly below the mean values, i.e. specific heat is 1672 J/(kg·K) while the corresponding mean value is 1678 J/(kg·K) and density is 210 kg/m3 while the corresponding mean value is 211 kg/m3. Somewhat smaller values were found for the observation #2. The observed variation of individual values of the measured properties was characterized by the standard deviation values of 7.7 J/(kg·K) and 5.3 kg/m3 for specific heat and density, respectively. Such a variation is small when considering bio-based materials usually revealing high diversification of structure and properties of the materials.

Comment #3. Could you carefully explain why individual observation values of thermal conductivity measured with Isomet 2114 varied greatly at the same temperature level (fig. 1).

Answer. The individual values of the thermal conductivity as measured with Isomet 2114 varied greatly at the same temperature level primary due to different surface roughness of the investigated material causing diverse contact resistance between the tested samples and the sensor of the meter. According to Hammerschmidt (2002) the imperfect contact significantly underrates the measured values of thermal conductivity. It is the most important disadvantage of the employed method in Isomet 2114 when using the meter for natural bio-based insulating materials.

Hammerschmidt, U. Guarded hot-plate (GHP) method: uncertainty assessment. Int. J. Thermophys. 2002, 23, 1551-1570. DOI: 10.1023/A:1020737900473

Comment #4. It would be more coherent to make a comparison between fig. 3 and fig. 4.

Answer. Such a comparison would be possible for the same locations of thermocouples and different materials. Unfortunately, it is not the case of the study.

Comment #5. Please present data of different thermocouples, such as thermocouple number #3 and #5, to make the table more comprehensive (table 3).

Answer. The values of the global in time relative error as presented in Table 3 were calculated for the thermocouples being mounted inside the examined material. The thermocouples #3 and #5 as well as the thermocouple #7 were always installed at the faces of samples. These thermocouples were used for registering data for the Dirichlet boundary condition at all faces of the samples during the transient heat transfer experiments. Therefore, the global in time relative error determination for such locations is rather not justified. As considers the positions of the thermocouples we checked it and corrected in the revised version of the paper.

Round 2

Reviewer 1 Report

The authors have addressed my questions.